# Triple-Negative Breast Cancer and the COVID-19 Pandemic: Clinical Management Perspectives and Potential Consequences of Infection

**DOI:** 10.3390/cancers13020296

**Published:** 2021-01-15

**Authors:** Justin M. Brown, Marie-Claire D. Wasson, Paola Marcato

**Affiliations:** 1Department of Pathology, Dalhousie University, Halifax, NS B3H 4R2, Canada; justin.brown@dal.ca (J.M.B.); mc.wasson@dal.ca (M.-C.D.W.); 2Department of Microbiology and Immunology, Dalhousie University, Halifax, NS B3H 4R2, Canada

**Keywords:** triple-negative breast cancer, metastasis, clinical management, SARS-CoV-2, COVID-19

## Abstract

**Simple Summary:**

The Coronavirus disease (COVID-19) pandemic has resulted in challenges to cancer management, exacerbated by limited clinical resources and caution in preventing COVID-19 transmission between patients and healthcare professionals. The neglect of breast cancer (in particular, triple-negative breast cancer (TNBC)) patients during the outbreak could negatively impact their overall survival, as delays in treatment and consultations provide vital time for tumor progression and metastasis. Herein, we review the shifting clinical management of TNBCs during the COVID-19 outbreak. The suggested treatment recommendations can hopefully minimize virus exposure without sacrificing patient care during times when healthcare systems are overburdened. Further, we review published RNA-seq data to assess the theoretical infectability of metastatic TNBCs to Severe Acute Respiratory Syndrome Coronavirus 2 (SARS-CoV-2) infection. These analyses highlight the potential of the virus to infect TNBC cells. Given the known increased susceptibility of cancer cells to viral infection, this additional host cell reservoir may make patients with metastatic disease particularly vulnerable to COVID-19 morbidities.

**Abstract:**

The COVID-19 pandemic has caused the need for prioritization strategies for breast cancer treatment, where patients with aggressive disease, such as triple-negative breast cancer (TNBC) are a high priority for clinical intervention. In this review, we summarize how COVID-19 has thus far impacted the management of TNBC and highlighted where more information is needed to hone shifting guidelines. Due to the immunocompromised state of most TNBC patients receiving treatment, TNBC management during the pandemic presents challenges beyond the constraints of overburdened healthcare systems. We conducted a literature search of treatment recommendations for both primary and targeted TNBC therapeutic strategies during the COVID-19 outbreak and noted changes to treatment timing and drugs of choice. Further, given that SARS-CoV-2 is a respiratory virus, which has systemic consequences, management of TNBC patients with metastatic versus localized disease has additional considerations during the COVID-19 pandemic. Published dataset gene expression analysis of critical SARS-CoV-2 cell entry proteins in TNBCs suggests that the virus could in theory infect metastasized TNBC cells it contacts. This may have unforeseen consequences in terms of both the dynamics of the resulting acute viral infection and the progression of the chronic metastatic disease. Undoubtedly, the results thus far suggest that more research is required to attain a full understanding of the direct and indirect clinical impacts of COVID-19 on TNBC patients.

## 1. Introduction

The coronavirus disease 2019 (COVID-19) pandemic caused by severe acute respiratory syndrome coronavirus 2 (SARS-CoV-2) has resulted in the extraordinary shifting prioritizations of healthcare resources. Global shortages of medical equipment, personal protective equipment and staff (due to self-isolation and personal illness) have shifted resource allocation to the most critical and urgent cases. While traditionally considered top priority, cancer screening and management have been pushed to the periphery of healthcare to allow increased time and space to be distributed to COVID-19 patients. Breast cancers comprise a significant portion of all human cancers, necessitating an investigation into the management of this heterogeneous group of diseases during the COVID-19 pandemic. Accordingly, a survey conducted by a group in the US reported that 44% of the 609 breast cancer patients included in the study experienced delays in their cancer treatments during the pandemic [1]. Additionally, the outbreak has resulted in significant reduction of clinical prevention for breast cancer patients, such as genetics counselling and breast screening [2], which will inevitably lead to an increased number of undiagnosed breast cancers. The COVID-19 pandemic has thus resulted in a bi-directional challenge to cancer treatment and management, exacerbated by limited clinical resources and caution in preventing COVID-19 transmission between patients and healthcare professionals.

The neglect of breast cancer patients during the outbreak could negatively impact their overall survival, as delays in surgery and patient-physician consultations provide vital time for tumor progression and metastasis. This is particularly true for triple-negative breast cancer (TNBC), an aggressive breast cancer subtype, with heightened risks of relapse and metastasis [3,4]. Some TNBC patients have experienced delays exceeding four months for their imaging tests during the COVID-19 pandemic [5]. Treatment postponement greater than six months is associated with up to a 20% decrease in breast cancer patient overall survival and even worse outcomes for TNBC patients [6]. Delays in treatment would certainly impose a greater risk of adverse clinical outcomes for patients with metastatic TNBC. Thus, our understanding of SARS-CoV-2 infection and COVID-19 disease in this unique group of patients is imperative. Taken together, these findings support the urgent need to devise treatment strategies for TNBC patients during the COVID-19 outbreak that minimize delays while ensuring the safety of patients and healthcare professionals.

Herein, we review the shifting clinical management of TNBCs during the COVID-19 outbreak. The suggested treatment recommendations can hopefully minimize SARS-CoV-2 exposure without sacrificing patient care during times when healthcare systems are overburdened. Further, we review published RNA-sequencing (RNA-seq) datasets to assess the theoretical infectability of metastatic TNBCs to SARS-CoV-2 infection based on critical host proteins involved in SARS-CoV-2 pathogenesis, angiotensin-converting enzyme 2 (ACE2) and transmembrane protease serine 2 (TMPRSS2). These analyses highlight the potential of the virus to infect metastatic TNBC cells. Given the known increased susceptibility of cancer cells to viral infection in general [7], this may make patients with metastatic disease particularly vulnerable to COVID-19 morbidities.

## 2. SARS-CoV-2

A member of Coronaviridae, SARS-CoV-2 originated in China, in 2019, giving rise to a global pandemic and unprecedented health and economic crisis. SARS-CoV-2 infections range clinically from asymptomatic infection or mild upper respiratory tract illness to severe viral pneumonia accompanied by respiratory failure and potential death [8,9]. The entry of SARS-CoV-2 into host cells is a critical process facilitating its widespread infection. Importantly, there are key cell entry mechanisms unique to SARS-CoV-2. The virus spike protein trimer binds the human ACE2 receptor with high binding affinity and is proteolytically activated by protease TMPRSS2 and lysosomal proteases (cathepsins), facilitating host cell entry [10].

The prevention of COVID-19 via vaccination and/or treatment of COVID-19 patients with anti-SARS-CoV-2 agents will be critical in patients with underlying medical conditions, such as those receiving treatment for cancer, a group with elevated risk of SARS-CoV-2 infection and worse COVID-19 disease outcomes. Recently, multiple vaccines have been approved globally for use against SARS-CoV-2 including BNT162b2 (Pfizer/BioNTech), mRNA-1273 (Moderna) and AZD1222 (Oxford-AstraZeneca). Immunization against COVID-19 is currently underway, where people of advanced age and/or those with a high risk of severe illness and death from COVID-19 have been identified as key populations for early immunization. Despite the initiation of COVID-19 immunization, the treatment guidelines for cancer patients described herein remain relevant and should be considered until the spread of SARS-CoV-2 is controlled and hospital resources are no longer at risk. In addition to vaccines, the anti-SARS-CoV-2 agent, Remdesivir (GS-5734), has been FDA approved for the treatment of adults and children over the age of 12 requiring hospitalization due to COVID-19. Remdesivir is an adenosine analogue that inhibits viral replication. Remdesivir was shown to reduce recovery time in adult patients hospitalized with COVID-19 who presented with lower respiratory tract infections (NCT04280705). Further research is required to assess the safety of Remdesivir for patients with cancer. Until anti-SARS-CoV-2 treatments and COVID-19 vaccines become widespread, the careful management of cancer remains a priority.

## 3. Cancer Management during the COVID-19 Pandemic

The treatment of patients with various cancers during the COVID-19 pandemic presents a unique set of challenges. Cancer patients or those with a history of cancer face a disproportionately higher risk of SARS-CoV-2 infection and a poor prognosis relative to COVID-19 patients without cancer [11,12,13,14]. However, many of these studies have been limited by low numbers of patients and a lack of diversity in geography, age, sex, race/ethnicity and cancer treatment information. The disparity in SARS-CoV-2 infections among cancer patients relative to non-cancer patients may be explained by evidence that cancer patients are immunocompromised while receiving anti-cancer treatments (i.e., surgery, chemotherapy, targeted therapies) or supporting medications (i.e., steroids) and experience the inherently immunosuppressive nature of their cancer [11,12,15]. Furthermore, cancer patients are often older than 60 years of age with one or more co-morbidities and are in frequent contact with healthcare services and facilities during and after cancer treatment, increasing their risk of SARS-CoV-2 infection and COVID-19 mortality [15]. Clinicians are now presented with the difficult task of healthcare restructuring that concurrently favors the reduction of COVID-19 transmission and the delivery of high-quality care for cancer patients. The high degree of heterogeneity among cancers and the broad spectrum of cancer progression profiles and variation in subtype-specific clinical implications mean that prioritizations for treatment must be made.

These studies have highlighted cancer patients as an important at-risk group for adverse clinical prognosis following SARS-CoV-2 infection. While more data on cancer patients with COVID-19 are required, a retooling of patient cancer treatment that reduces the risk of SARS-CoV-2 infection without sacrificing potentially life-saving cancer treatments is favorable. An abundance of literature has been published focusing on the clinical outlook of cancer patients with COVID-19. However, less information exists on specific cancers and clinically relevant cancer subtypes. Thus, more research will be necessary to fully understand the effects of COVID-19 on these patients and to inform cancer type-specific treatment guidelines, as is the case for TNBC.

## 4. TNBC Management during COVID-19

For the effective prioritization of treatment for breast cancer patients not suspected to have COVID-19 during the pandemic, Dietz and colleagues have suggested the categorization of breast cancer patients into three priority levels (A, B & C) [16]. They also provide treatment considerations and recommendations specific to the COVID-19 pandemic for each level. Priority A patients are considered to have an immediately life-threatening or symptomatic condition requiring urgent clinical intervention, while priority B patients have conditions that require treatment prior to the conclusion of the pandemic. The treatment of priority C patients can be postponed pending the end of the pandemic. Most breast cancer patients will fit into the priority B category, which is further subdivided into groups according to priority (B1–B3, where B1 patients are the highest priority). Given the aggressiveness of their disease, patients with TNBC are often among the highest priority for treatment. In accordance, TNBC patients are category B1 along with patients with HER2+ breast cancers.

To explore changes in TNBC clinical strategies induced by the COVID-19 pandemic, we conducted a literature search to provide a detailed summary of suggested TNBC treatment strategies in response to COVID-19. We queried NCBI PubMed with the keywords “breast cancer” and “COVID-19” which returned 118 articles. From these, we eliminated articles with titles lacking our search terms. For these papers, we were solely focused on retrieving information pertaining to TNBC. We summarize our findings in the “Traditional TNBC therapies” section and emphasize chemotherapy-related recommendations in Table 1.

## 5. Traditional TNBC Therapies

TNBC is traditionally treated with a combination of chemotherapy, radiation and surgery. The pandemic has resulted in numerous challenges to chemotherapy administration, most notably, the high frequency of hospital visits required and its immunosuppressive nature. To minimize possible exposure to the virus, it has been suggested that patients receive chemotherapy in wide intervals. For example, weekly paclitaxel administration is preferred over thrice-weekly docetaxel [33]. Outpatient administration of chemotherapy has also been raised as a potential solution to reduce hospital visits. For example, there are chemotherapy options that do not require patients to receive treatments in healthcare centers. For example, oral chemotherapy, such as capecitabine, may be taken for TNBC patients after neoadjuvant chemotherapies if residual disease persists. Further, low-dose capecitabine taken after surgery and post-surgery chemotherapy for one year among early-stage TNBC patients has been shown to diminish the risk of disease recurrence (NCT01112826) [34]. Another clinical trial showed that the addition of capecitabine to standard chemotherapy after surgery improved disease-free survival among early-stage TNBC patients with an absence of adverse side effects [35]. Thus, oral capecitabine may be taken by early-stage TNBC patients following either neoadjuvant chemotherapy or surgery and post-surgery chemotherapy to reduce patient contact with the healthcare system.

The use of neoadjuvant chemotherapy may serve as an alternative treatment route in the case of delayed or cancelled surgeries to limit the use of operating room resources [36]. Upon collaborative assessment of patient comorbidities and risk factors, neoadjuvant chemotherapies may be administered in place of surgery for TNBC patients as beginning with surgery is not required in the presence of systemic therapy. The standard of care for TNBCs already includes neoadjuvant chemotherapy to which patients respond well, potentially facilitating effective disease management during surgery deferral [37]. In addition, a study reported that surgery deferral for up to 6–8 weeks following chemotherapy did not impact survival outcomes of breast cancer patients [33,38], supporting the initial use of neoadjuvant chemotherapies among TNBC patients. Depending on patient and institutional factors, however, neoadjuvant therapy may not need to precede surgery. Spring et al. suggests that neoadjuvant chemotherapy is a viable option for tumors greater than 0.5 cm^3^ [39]. For small tumors, Spring et al. recommend surgery as the favored treatment strategy as the risk of TNBC tumor progression outweighs the need for surgical resources and risk of COVID-19 contraction [39]. Patients who have completed neoadjuvant treatment should undergo surgery within 4–8 weeks, allowing for recovery from toxicity and myelosuppression [27,28]. TNBC patients unresponsive to primary systemic therapies should receive surgery in less than four weeks, depending on tumor burden [20].

Breast conserving surgery is favored over a full mastectomy where possible if radiation oncology services are available and deferrable and frequent hospital visits are acceptable [16]. Studies assessing the risk of COVID-19 transmission following cancer surgery in the UK determined that the risk of COVID-19 exposure following surgery is minimal when staff and patients follow proper precautions [40,41]. No patient who underwent surgery in the study period tested positive for COVID-19, and there were no critical care admissions or mortality events [40,41]. These reports suggest that with proper precautions, essential breast surgeries can occur safely, and non-critical surgeries can be resumed during the pandemic for high-risk patients, such as TNBC patients.

Many patients opt to undergo breast reconstruction following resection, which has shown to improve how patients cope with their diagnosis and cancer journey, solidifying breast reconstruction as a fundamental facet of breast cancer treatment. A group in Italy devised a protocol for breast reconstruction surgery that minimized the risk COVID-19 exposure to both physicians and the patient as well as accelerating the post-operative recovery and hospital discharge [42]. At the date of publication, 51 patients had undergone breast reconstruction following this new protocol with no additional postoperative complications and no increased risk of COVID-19 infection to the patients or staff, supporting the notion that patients can undergo reconstructive surgery in the COVID-19 era.

Radiotherapy, as an alternative or in addition to chemotherapy and surgery, is used extensively in the treatment of breast cancer. Radiotherapy offers several advantages over other cancer treatment options during the COVID-19 outbreak. First, it does not require use of in-demand resources such as respirators or intensive-care unit beds (unlike surgery) [43]. Second, the radiotherapy administration schedule can be tailored to reduce the number of hospital visits [43]. Third, radiation does not subject the patient to the same extent of immunosuppressive effects as chemotherapy [43]. Taken together, these points suggest that radiotherapy may be an ideal choice when appropriate for TNBC treatment during the pandemic if proper PPE is donned and care is taken to limit the spread of disease between healthcare professionals and patients.

If the need to limit radiotherapy arises due to resource constraints, Braunstein and colleagues suggest a prioritization scheme for breast cancer patients where TNBC patients are deemed high or mid-priority for radiation access based on the node status of the patient [44]. Following neoadjuvant or adjuvant chemotherapy and surgery, stage I to III TNBC patients should be referred for radiation oncology according to standard treatment guidelines [10,45]. Patients who have completed neoadjuvant therapy who lack therapeutic targets (i.e., TNBC) are high priority for preoperative radiation therapy [33]. Despite the COVID-19 pandemic, delays in radiation are not advised due to the risk of recurrence [10,46]. It is recommended that radiation therapy be administered within 16 weeks following the conclusion of chemotherapy or surgery [27]. TNBC patients are also at a high priority for adjuvant radiation therapy, which should be administered 2–4 months post-surgery [4,17,21]. The effectiveness of radiation therapy as a primary TNBC treatment, rather than in an adjuvant setting, is still debated. Evidence suggests that TNBC is resistant to radiation when used alone; however, radiotherapy with poly ADP-ribose polymerase (PARP) inhibitors may be effective due to their synergistic DNA damage-enhancing effects [47].

## 6. Targeted TNBC Therapies

A recent report by the national health service (NHS) classified cancer patients receiving treatments including targeted anti-PD-1/PD-L1 and PARP inhibitors are at an elevated risk of SARS-CoV-2 infection [48]. Hence, the administration of such targeted therapies, which occurs in combination with traditional therapies in TNBC patients, presents a unique set of COVID-19- related challenges that pertains to this subtype.

Anti-PD-L1 therapy atezolizumab (ATZ) plus paclitaxel is an approved therapy for patients with PD-L1 positive metastatic TNBC [32]. Fortunately, immune checkpoint inhibitors such as ATZ do not exert immunosuppressive effects and thus should not augment the risk of TNBC of SARS-CoV-2 infection due to greater immunosuppression [49,50]. Strikingly, immune checkpoint inhibitors may even boost pathogen-specific immune responses, and cases of viral or bacterial infections following treatment with ICIs are limited [51,52]. However, there are other side effects of PD-1/PD-L1 inhibition which may exacerbate the risk of SARS-CoV-2 infection and adverse outcomes [53]. For example, cytokine release syndrome is a systemic inflammatory disease characterized by the release of circulating inflammatory cytokines such as interleukin-6 (IL-6) and interferon gamma [54]. Conversely, it has been suggested that this cytokine storm may be beneficial for patients with TNBC who are receiving anti-PD-1/PD-L1 therapy and chemotherapy [55]. While the cytokine storm is associated with acute respiratory distress in COVID-19 patients, it is suggested that TNBC patients with isolated lung metastases receiving combination immunotherapy may benefit from the COVID-19-associated cytokine storm in the lungs which may also act on metastatic lung nodules [55,56].

Many TNBCs exhibit a deficiency in homologous recombination and the associated repair of double strand DNA breaks, which makes them sensitive to PARP inhibitors. Intriguingly, there is evidence to support the use of PARP inhibitors to hamper SARS-CoV-2 infection and associated severe COVID-19 disease [57]. Specifically, PARP inhibition may function against COVID-19 by limiting macrophage overactivation and associated cytokine storm along with protection against cell death. PARP inhibition reduces the levels of inflammatory cytokines associated with SARS-CoV-2-mediated cytokine storms. Thus, TNBC patients with concomitant COVID-19 receiving PARP inhibitors may experience better outcomes.

## 7. Specific Implications for Metastatic TNBC during the COVID-19 Pandemic

Within five years of their diagnosis, women with TNBC are more likely to experience metastasis relative to women with other breast cancer subtypes [58]. Patients with TNBCs experience metastasis the bone, lungs, liver and brain. Therefore, in the context of management of TNBC during the COVID-19 pandemic, there are likely different factors to be considered if the TNBC is local and surgically removed versus a patient with metastatic disease.

To our knowledge, the mortality risk of lung metastasis in TNBC patients has not yet been assessed in the context of COVID-19. However, a recent study found that both lung cancer patients or patients with lung metastasis and COVID-19 were at a heightened risk of death compared to non-metastatic cancer patients with COVID-19 [13]. Based on this finding, TNBC patients with lung metastasis likely have elevated mortality risk, although the specific implications have yet to be determined.

Additionally, the elevated risk of lung metastasis in TNBC complicates the diagnosis from respiratory-related symptoms in this patient cohort. In a case study reported by Chen and Li, the presence of a cough in an advanced gynecological cancer patient was initially perceived as a sign of lung spread [59]. Follow-up analyses determined that the patient was positive for COVID-19, attributing the source of the respiratory-associated symptoms to the virus rather than lung metastasis [59]. Other reports state that pleural metastasis can mimic COVID-19 symptoms, further complicating the source of suspicious symptoms [60]. COVID-19 testing in cancer patients at multiple points in their care is thus critical for accurate disease tracking and management.

The implications of SARS-CoV-2 infection for patients with metastatic TNBCs are multifaceted and complex. SARS-CoV-2 infection in an environment where lung function is already reduced may significantly worsen the risk of COVID-19 mortality. COVID-19 patients requiring hospitalization often require ventilation and experience adverse outcomes, which is associated with viral load. Importantly, elevated viral load has been associated with advanced age, comorbidities and recent chemotherapy, all of which are associated with cancer. Further, intubation and mortality following hospitalization were highest in patients with the greatest viral loads [61]. The SARS-CoV-2 viral load is also associated with substantially increased IL-6 levels, indicative of cytokine storm and poor outcomes [62].

Metastasized TNBC cells may represent a cell population with enhanced susceptibility to viral infection and replication. In line with this, during the multi-step process whereby normal cells become cancerous, among other “hallmark” acquisitions contributing to their oncogenic phenotype, cells become capable of evasion of host immune-mediated recognition and destruction [63]. Specifically, cancer cells acquire defective anti-viral defenses, including pathways mediated by interferons, increasing their susceptibility to viral infection [7]. Thus, assuming adequate cellular expression of ACE2 and TMPRSS2, patients with metastatic TNBC represent a patient population with potential enhanced vulnerability to SARS-CoV-2 infection and poor outcomes due to widespread viral infection facilitated by cancer metastasis.

To determine whether TNBCs are possibly susceptible to SARS-CoV-2 infection, we interrogated published RNA-seq datasets to analyze the expression of both ACE2 and TMPRSS2 in TNBC patient tumors (primary tumors and metastatic) in comparison to cell lines known to be highly infectable by SARS-CoV-2 relative to a panel of known housekeeping/reference genes (Figure 1) [64]. These analyses showed that TNBC patient tumors and metastasized breast cancers exhibited expression of both ACE2 and TMPRSS2 at least within the range of the infectable cell lines, suggesting that TNBC cells that come in contact with SARS-CoV-2 may be susceptible to infection. Consistent with these findings, analysis of ACE2 and TMPRSS2 expression across a panel of metastatic tumors also showed that both genes are expressed in metastatic breast tumors (Figure 2). This further illustrates the potential for SARS-CoV-2 infection of metastatic breast cancer cells which may lead to heightened viral load and associated morbidities (Figure 3) [61,62]. Alternatively, it is also important to mention possible effects that SARS-CoV-2 infection may have on cancer cells present in the lung, due to the enhanced susceptibility of cancer cells to viral infection (Figure 3). Future case reports and studies may reveal that metastatic cancer patients that were infected with SARS-CoV-2 experience discernable improvements in the tumor burden.

## 8. Conclusions and Future Directions

Treating TNBC is a clinically challenging endeavor due to lack of targetable receptors and high propensity for metastasis. The COVID-19 pandemic has exacerbated these obstacles by restricting and delaying treatment options for patients. While therapeutic strategies for breast cancer management have been altered during the pandemic, the goal remains to maintain the standard of breast cancer care.

Here, we have provided a summary of shifting clinical management of TNBC during the pandemic. This review suggests possible treatment avenues for TNBC patients to better prepare clinicians to make informed treatment strategies for TNBC management. However, caution should be taken when implementing these recommendations as more research is needed to understand the impacts of these shifts in TNBC patient outcomes.

Given the global and unprecedented nature of the COVID-19 pandemic, we have never faced such an urgent need to prioritize healthcare resources and the treatment of patients with COVID-19 while maintaining a high standard of care for patients with other pathologies. This has resulted in limited knowledge of SARS-CoV-2 infection and transmission in healthy individuals and those with pre-existing medical conditions, such as cancer. The impact of SARS-CoV-2 on patients with aggressive cancers (i.e., TNBC) is especially enigmatic and requires urgent research. Our analyses of published datasets (Figure 1 and Figure 2) suggest that metastatic TNBC could be infected by SARS-CoV-2 based on ACE2 and TMPRSS2 expression (Figure 3). This needs to be validated in laboratory models and the consequence of such infection determined.

The need for a better understanding of COVID-19 has highlighted the need for open-access publication of SARS-CoV-2/COVID-19 patient data to facilitate novel research and global collaboration pertaining to the impacts of COVID-19 on other diseases, such as cancer. For example, access to lung tissue gene expression data from TNBC patients would allow the investigation of possible TNBC-specific host protein and SARS-CoV-2 protein interactions and their significance in SARS-CoV-2 susceptibility and potential treatment avenues among TNBC patients [68].

While COVID-19 has posed enormous pressure on our healthcare systems, this global event has forced a shift in management protocols for breast cancer. Following the peak of the pandemic, these solutions may result in more efficient treatment of breast cancer without compromising the quality of patient care. This review provides a starting point for further research investigating TNBC and COVID-19. Such research is paramount during the COVID-19 crisis to ensure a high standard of care, preventing adverse TNBC patient outcomes induced by COVID-19 disease and/or obstacles to effective cancer treatment. Finally, we call for the timely development of guidelines for the management of patients with metastatic TNBC during the COVID-19 pandemic to efficaciously reduce the risk of SARS-CoV-2 infection and select the most advantageous clinical treatment avenues.

## Figures and Tables

**Figure 1 cancers-13-00296-f001:**
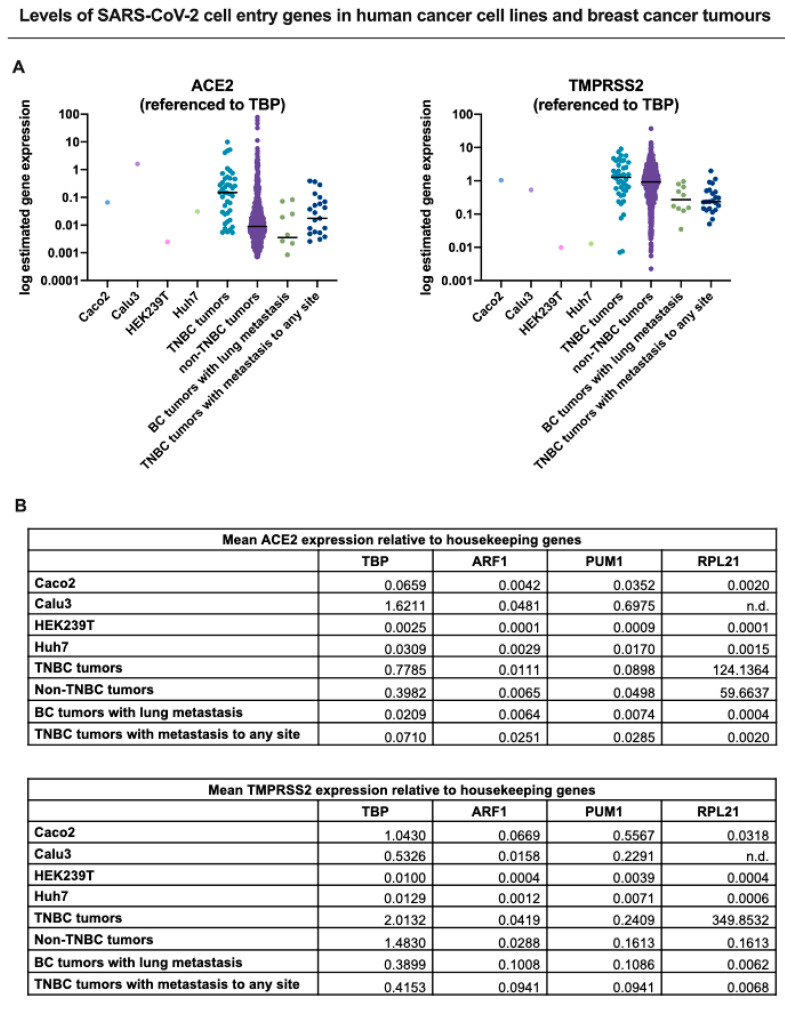
ACE2 and TMPRSS2 expression in breast cancer is comparable to cell lines permissive to SARS-CoV-2 infection. To compare expression of ACE2 and TMPRSS2 across datasets, we normalized the data to four different reference genes and depicted these as dot plots in (**A**) to one reference gene (*TBP*) and summarized in table format in (**B**) for all four reference genes (*TBP, ARF1*, *PUM1, RPL21*). TPM normalized gene expression for ACE2 and TMPRSS2 in four human cell lines permissive to COVID-19 infection and replication: Caco-2 (colorectal adenocarcinoma cells), Calu-3 (lung cancer cells), HEK239T (embryonic kidney 293 cells), Huh7 (hepatoma cells), were calculated from raw RNA-seq counts obtained from NCBI’s Gene Expression Omnibus (*GEO*) Datasets database (GSE140066, GSE147507, GSE137556, GSE129277, respectively). Only the control/non-treated samples from each dataset were included in the above analysis. RNA Seq V2 RSEM normalized gene expression for TNBC and non-TNBC tumours was obtained through the Breast Invasive Carcinoma (TCGA, Cell 2015) dataset accessed through cBioPortal [65,66]. All metastatic tumor data (breast cancer (BC) tumors with lung metastasis, metastatic TNBC tumors) were acquired through The Metastatic Breast Cancer Project (Provisional, February 2020) retrieved using cBioPortal. The results for the metastatic dataset were normalized by RNA Seq V2 RSEM.). The results presented in Figure 1 were derived in part from data generated by The Cancer Genome Atlas (TCGA) research network. https://www.cancer.gov/tcga.

**Figure 2 cancers-13-00296-f002:**
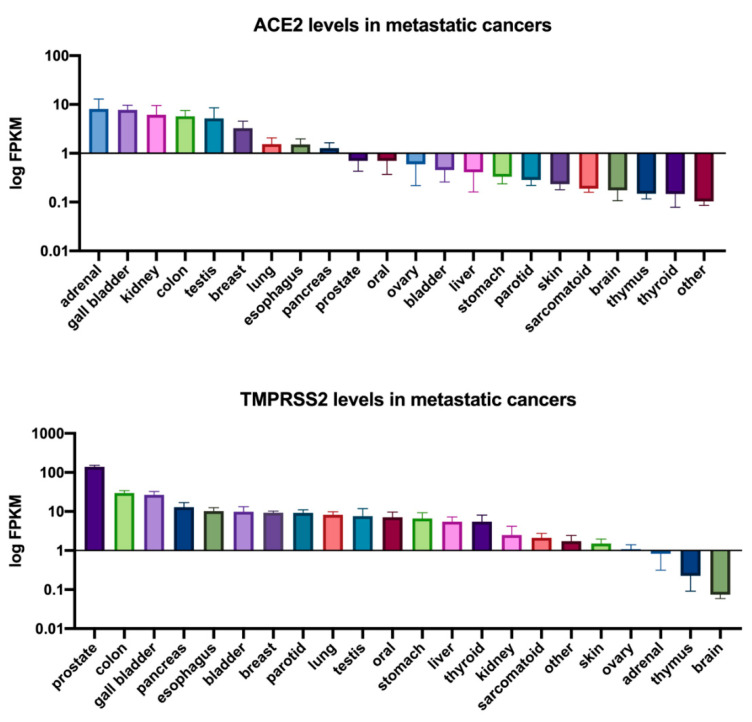
ACE2 and TMPRSS2 expression in metastatic cancers. ACE2 and TMPRSS2 FPKM expression values for metastatic cancers were obtained from the MET500 dataset [67] accessed through xenabrowser.net. The mean expression of ACE2 or TMPRSS2 for each tumor site is reported with the standard error of the mean (SEM). The results presented in Figure 2 were derived in part from data generated by The Cancer Genome Atlas (TCGA) research network. https://www.cancer.gov/tcga

**Figure 3 cancers-13-00296-f003:**
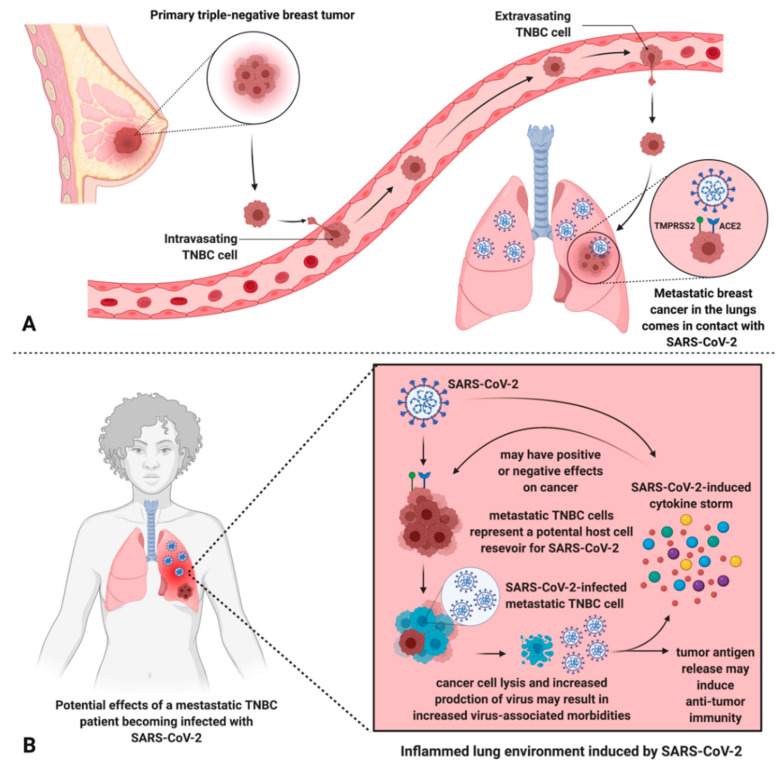
Metastatic TNBC cells in the lungs may be infected by SARS-CoV-2, leading to unknown effects on the patient’s cancer and potential amplification of viral infection/morbidities. (**A**) TNBC metastasized to the lungs theoretically permits the interaction between SARS-CoV-2 and TNBC cells via ACE2 and TMPRSS2, exploited by SARS-CoV-2 for host cell entry. (**B**) An inflamed lung environment induced by SARS-CoV-2 results in a cytokine storm which leads to unknown effects on metastatic cells but may increase virus-associated morbidities. Further, metastatic breast cancer cells may represent an additional host cell SARS-CoV-2 reservoir, leading to increased viral load and virus-associated morbidities, and unknown consequences on the progression of the cancer. Tumor cell lysis can also liberate tumor antigens, inducing anti-tumor immune responses.

**Table 1 cancers-13-00296-t001:** Chemotherapy and targeted therapy recommendations for TNBC patients during COVID-19.

Treatment Type	Treatment Recommendations
General Recommendations for Chemotherapy	Chemotherapy is a high priority for TNBC patients, even in the event of an overrun healthcare system [4,17,18,19]. Neoadjuvant and adjuvant chemotherapy are recommended for early stage TNBC patients [5,17,20,21]. Chemotherapy should be administered within eight weeks of initial diagnosis [22]. Oral chemotherapeutic agents and those with lower risk of immunosuppression, such as capecitabine or vinorelbine, are preferred to other methods requiring hospital visits [18]. A cost-benefit analysis should be performed when deciding on the composition of chemotherapy regimens considering the risk of haematological toxicity and immunosuppression and patient age and comorbidities in the context of the COVID-19 pandemic [18,23]. To reduce patient contact with the healthcare system, chemotherapy should be administered 3- or 4-weekly as opposed to weekly [19,24]. Is it not recommended that older patients with early stage TNBC receive chemotherapy due to the heightened risk of complications if SARS-CoV-2 infection ensues and the limited therapeutic benefits. A modified treatment strategy minimizing immunosuppression should instead be considered for these patients (i.e., ≥70 years old) [19,25].
Neoadjuvant Recommendations	Neoadjuvant therapy is recommended for TNBC due to its high rate of clinical success in moderating tumor growth prior to surgery [18]. Post-neoadjuvant therapy is recommended for patients not achieving a pathological complete response following neoadjuvant treatment, enhancing disease management and survival [17]. Neoadjuvant therapy can be used to delay surgery during the pandemic given the availability of hospital resources [26].
Adjuvant Recommendations	Adjuvant therapy should start no later than two months following surgery due to the increased risk of recurrence and death associated with delays [19]. Adjuvant capecitabine can be used to treat TNBC patients with residual disease who have undergone neoadjuvant chemotherapy [19,24,27,28]. Adjuvant therapy can be combined with targeted therapies for locally advanced TNBC patients [29].
Considerations for Metastatic TNBC	Chemotherapy regimens should minimize immunosuppression. For example, monotherapy should be favored over combination treatments to reduce myelosuppression; prophylactic CSF may reduce neutropenia; and the use of corticosteroids should be limited considering chemotherapy regimen [19]. Oral treatment should be prioritized. Patients treated with anthracyclines and taxanes prior to the pandemic may benefit from transition to oral capecitabine or metronomic cyclophosphamide [19,24,30,31]. For inoperable or metastatic TNBCs, consider Poly ADP-ribose polymerase (PARP) inhibitors (i.e., olaparib or talazoparib) if patients have a *BRCA* mutation [19,24]. Despite the low frequency of olaparib-associated pneumonitis, it, along with the potential for myelosuppression should be considered [19]. A combination of chemotherapy and the PD-L1 inhibitor atezolizumab should be considered for advanced or metastatic TNBC [4,17], especially given the comparatively lower risk of hematological complications relative to standard chemotherapy [32].

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
