# Peer review of "Triple-Negative Breast Cancer and the COVID-19 Pandemic: Clinical Management Perspectives and Potential Consequences of Infection"

_cancers, 2021, doi:10.3390/cancers13020296_

Round 1

Reviewer 1 Report

The authors nicely summarize the problem of TNBC, which is the most aggresive subtype of breast cancer and in most cases requires immediate action regardless of the Covid 19. The chemotherapy and targeted therapy recommendations  for TNBC patients during COVID 19 pandemiia are meaningful and useful.

In addition, the review article provides insights into the interwining of cancer cells in TNBC and SARSCov2 virus, via angiotensin-converting enzyme 2 and transmembrane protease serine2.

It is also very important that it suggests new possibilities for investigating the interactions of  breast cancer and SARS-Cov-2 infection.

Author Response

Thank you!

Reviewer 2 Report

Thank you for giving me the opportunity to review this paper. This is an excellent and timely review. Maybe before publication, authors could update information like ''While there are currently no clinically approved preventative SARS-CoV-2 vaccines or antiviral drugs for the treatment of COVID-19, several are under investigation in clinical trials. As of November 2020.....''

Also, if the editor feels like, the recommended treatment regimes could be double-checked by a practicing clinician (Oncologist). 

Scientifically this is a sound review. ‘’ TNBC patient tumors and metastasized breast cancers exhibited expression of both ACE2 and TMPRSS2 at least within the range of the infectable cell lines, suggesting that TNBC cells that come in contact with SARS-CoV-2 may be susceptible to infection’’, the analysis of the ACE2 and TMPRSS2 is somewhat circumstantial but never the less it gives some indication for future experiments to confirm these finding.  

Author Response

Thank you!

As per the Reviewer's comment, we have updated the section discussing COVID vaccines and treatments.

Specifically, that paragraph now reads: 

The prevention of COVID-19 via vaccination and/or treatment of COVID-19 patients with anti-SARS-CoV-2 agents will be critical in patients with underlying medical conditions, such as those receiving treatment for cancer, a group with elevated risk of SARS-CoV-2 infection and worse COVID-19 disease outcomes. Recently, multiple vaccines have been approved globally for use against SARS-CoV-2 including BNT162b2 (Pfizer/BioNTech), mRNA-1273 (Moderna) and AZD1222 (Oxford-AstraZeneca). Immunization against COVID-19 is currently underway, where people of advanced age and/or those with a high risk of severe illness and death from COVID-19 have been identified as key populations for early immunization. Despite the initiation of COVID-19 immunization, the treatment guidelines for cancer patients described herein remain relevant and should be considered until the spread of SARS-CoV-2 is controlled and hospital resources are no longer at risk. In addition to vaccines, the anti-SARS-CoV-2 agent, Remdesivir (GS-5734), has been FDA approved for the treatment of adults and children over the age of 12 requiring hospitalization due to COVID-19. Remdesivir is an adenosine analogue that inhibits viral replication. Remdesivir was shown to reduce recovery time in adult patients hospitalized with COVID-19 who presented with lower respiratory tract infections (NCT04280705). Further research is required to assess the safety of Remdesivir for patients with cancers. Until anti-SARS-CoV-2 treatments and COVID-19 vaccines become widespread, the careful management of cancer remains a priority.

Reviewer 3 Report

This article presents a review of recent literature on the impact of the COVID19 pandemic on breast cancer management and particularly of TNBC patients. Particularly it addresses the issues that COVID19 infection may raise in case of metastatic disease and particularly lung metastasis. Indeed, patients with lung metastasis may present symptoms resembling those of COVID19 infection. It also makes an interesting transcriptome analysis of breast cancer cell lines and clinical samples showing a rather elevated level of the ACE2 and TMPRSS2 genes which act as receptors to the COVID19 virus in case of infection. This suggests that metastatic TNBC cells could be infected and potentially reduce the tumor burden. 

In conclusion this review article raises important issues for the medical oncology community in charge of TNBC patient management in this period of active COVID19 circulation and broaden perspectives. 

Author Response

Thank you!